# On-the-Go Vis-NIR Spectroscopy for Field-Scale Spatial-Temporal Monitoring of Soil Organic Carbon

**Javier Reyes** [1] and **Mareike Ließ** [1,2,*]

1 Helmholtz Centre for Environmental Research—UFZ, Department Soil System Science, 01620 Halle, Germany
2 Department of Agriculture, Food, and Nutrition, Data Science Division, University of Applied Sciences Weihenstephan-Triesdorf, 91746 Weidenbach, Germany
* Correspondence: mareike.liess@hswt.de

**Abstract:** Agricultural soils serve as crucial storage sites for soil organic carbon (SOC). Their appropriate management is pivotal for mitigating climate change. Continuous monitoring is imperative to evaluate spatial and temporal changes in SOC within agricultural fields. In-field datasets of Vis-NIR soil spectra were collected on a long-term experimental site using an on-the-go spectrophotometer. Data processing for continuous SOC prediction involves a two-step modeling approach. In Step 1, a partial least square (PLSR) regression model is trained to establish a relationship between the SOC content and spectral information, including spectral preprocessing. In Step 2, the predicted SOC content obtained from the PLSR models is interpolated using ordinary kriging. Among the tested spectral preprocessing techniques and semivariogram models, Savitzky–Golay and the Gap-Segment derivative preprocessing along with a Gaussian semivariogram model, yielded the best performance resulting in a root mean square error of 1.24 and 1.26 g kg$^{-1}$. A striping effect due to the transect-based data collection was addressed by testing the effectiveness of extending the spatial separation distance, employing data aggregation, and defining the distribution based on treatment plots using block kriging. Overall, the results highlight the high potential of on-the-go spectral Vis-NIR data for field-scale spatial-temporal monitoring of SOC.

**Keywords:** soil organic carbon; Vis-NIR spectroscopy; monitoring; pedometrics





## 1. Introduction

The spatial-temporal monitoring of soil organic carbon (SOC) in agricultural lands is instrumental in enabling climate-responsive agricultural management with a focus on enhancing SOC stocks. One of the key global frameworks that recognize the importance of SOC sequestration is the Paris COP 21 Climate Change Agreement [1], which emphasizes the critical role that soils play in capturing carbon dioxide from the atmosphere. By harnessing high-resolution spatial-temporal SOC data, it becomes possible to align agricultural practices with the goals outlined in this landmark agreement [2]. In essence, monitoring SOC at the field scale permits farmers to engage in precision carbon farming, adapt to climate variability, and contribute to global climate goals. The integration of advanced technologies like soil sensing is essential in collecting these data. And collaboration among scientists, policymakers, and farmers can foster the development and adoption of practices that maximize SOC sequestration in line with the objectives to combat climate change.

Conventional soil laboratory analysis often goes along with high costs, making it an unfeasible approach for extensive data acquisition [3]. To address this, sensor data are often incorporated alongside conventional data collection methods for constructing high-resolution field maps. One particularly promising avenue in this regard is proximal sensing applying Vis-NIR soil spectroscopy. Proximal sensing refers to measurements in direct contact with or close to the soil. It is recognized as a cost-effective method that can generate high-density data at field scale [4]. Soil spectroscopy entails the analysis of how soils

interact with electromagnetic radiation. It has been researched for its potential in predicting SOC among other properties [3,5,6]. Although the application of soil spectroscopy under field conditions is an area that is yet to be thoroughly explored, some studies demonstrate its potential through measurements at individual points [7–9]. Moreover, there is a growing interest in the utilization of on-the-go sensing for soil spectroscopy, which has proven to be more appropriate for high-resolution spatially continuous predictions at field scale [10–12].

Soil spectroscopy has emerged as a potent tool for monitoring SOC due to its ability to detect and analyze the interactions between electromagnetic radiation and soil constituents. In the Vis-NIR range (350–2500 nm), soil spectra exhibit weak overtones and combinations of fundamental vibrations caused by the bending and stretching of various soil compounds. This range is especially sensitive to organic matter, making it viable for SOC estimation [13]. As of now, the state of the art in soil spectroscopy predominantly involves laboratory-based analyses. Under controlled conditions, processed soil samples are subjected to spectral measurements, and the data are analyzed to establish relationships between spectral characteristics and SOC content. Among the methods employed, partial least squares regression (PLSR) has been widely used. It efficiently handles the spectra by extracting the information that is most relevant to SOC, and thus enables the development of robust prediction models [14].

Attempts have been made to extend soil spectroscopy to field conditions for continuous, field-scale SOC monitoring. This involves various approaches, such as UAV-based, airborne remote sensing, and proximal on-the-go sensing. For instance, unmanned aerial vehicles (UAVs) can carry sensors that capture soil spectra over large areas. Similarly, airborne remote sensing platforms can provide spectral data at broader scales. However, these methods can be limited by spectral resolution, image quality, and frequency of acquisitions [15,16]. Proximal on-the-go sensing represents a more direct approach, where sensors are close to the soil and can capture high-density data. However, implementing soil spectroscopy in the field is inherently challenging due to various environmental factors affecting the spectra, such as soil moisture, surface roughness, crop residuals and/or roots, incident light, soil texture, bulk density, and soil structure [17–19]. The resulting uncertainties in the SOC Vis-NIR relationship models can pose limitations for field mapping and the transferability of the models to other sites [20].

Long-term field experiments (LTEs) are designed to evaluate the long-term effects of various agricultural management practices on soil properties and crop traits [21]. By observing the influence of different practices on soil within the same experimental framework, LTEs provide time series data that are crucial for understanding how SOC levels change over time under different agricultural management practices. This makes LTEs an invaluable resource for making informed decisions about sustainable agricultural management with an emphasis on the vital role of SOC in soil health.

This study aims to delve into the feasibility and potential of employing on-the-go Vis-NIR spectroscopy as a tool for spatial-temporal monitoring of SOC. A comprehensive modeling procedure is formulated and presented as a central component of this investigation. Additionally, the study meticulously examines the specific influence that each stage of the modeling procedure has on predictive uncertainty. Through this multifaceted approach, the study seeks to shed light on the capabilities and limitations of on-the-go Vis-NIR spectroscopy in accurately capturing the spatial and temporal variations in SOC.

## 2. Materials and Methods

### 2.1. Study Area

Data were collected on the LTE site Static Fertilization Experiment in Bad Lauchstädt, Saxony-Anhalt, Germany (51°24′ N, 11°53′ E, 113 m a.s.l). The site is characterized by an average total annual precipitation of 470–540 mm and an average annual temperature of 8.5–9.0 °C. The soil was described as Haplic Chernozem developed from loess [22]. Accordingly, it has a topsoil texture varying between highly clayey silt (Ut4) and highly silty clay (Tu4) according to the German soil survey system [23]. The field experiment

was initialized in 1902 by Schneidewind and Gröbler on an area of c. 4 ha [24] with eight subfields (Figure 1A). From the initial crop rotation of winter wheat, sugar beet, summer barley, and potato, the root crops were replaced by silage maize from 2015 onwards. Different crops in nearby fields started the agricultural rotation, ensuring that all crops are always produced concurrently on the experimental site. Thirty dt of lime is applied to subfield 1 every 4 years in the spring. On subfield 8, legumes have been a part of the agricultural rotation every seventh and eighth year since 1926. The 288 plots as a whole vary according to how they were fertilized with minerals and organic fertilizer. One-third of each field is covered with farmyard manure applied at rates of 20 and 30 t ha$^{-1}$, respectively, while the other third is left devoid of organic fertilizer. Mineral fertilizer is applied in various N, P, and K combinations. In 1978, the experimental site's subfields 4 and 5 were modified to examine additional fertilizer treatments involving varied levels of N in combination with an adapted organic fertilizer treatment. Körschens and Pfefferkorn [25] delve into greater detail.

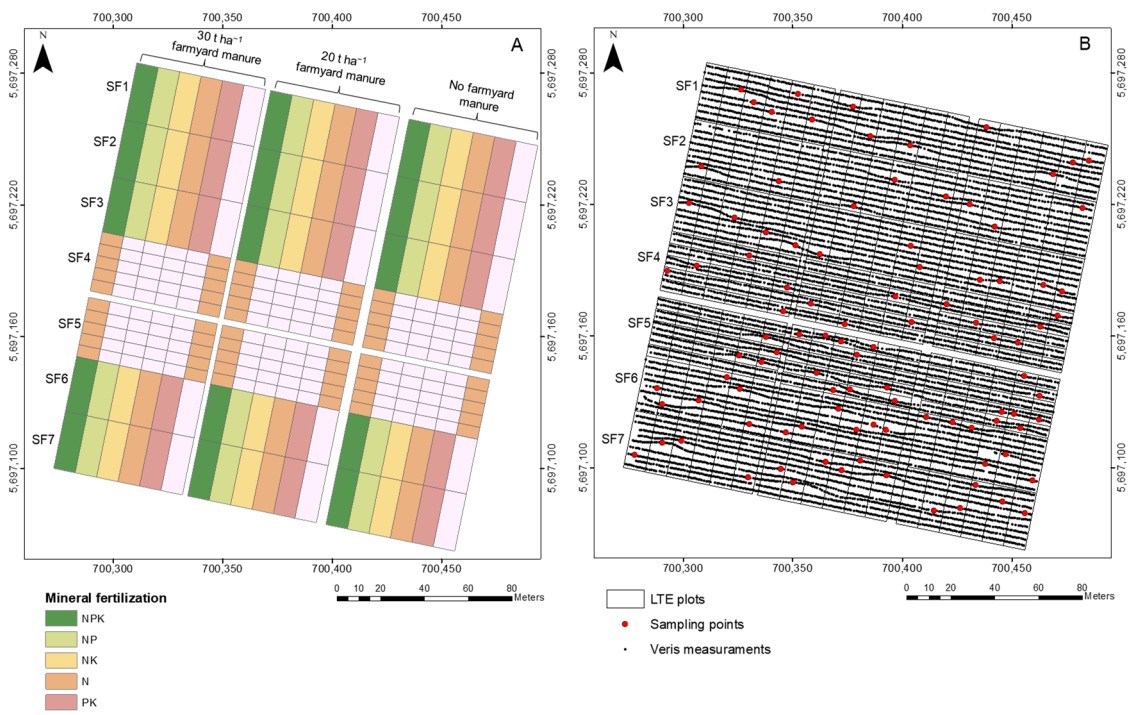

**Figure 1.** The study area located in Bad Lauchstädt. (**A**) Management factors of the long-term experiment. (**B**) Long-term experimental site with sampling points and Veris transects. Coordinate reference system: EPSG 25833.

## 2.2. Data Collection

In September 2018, soil samples were collected from 100 different locations at depths ranging from 0 to 10 cm (Figure 1B). To cover the spatial soil variability according to the LTE agricultural treatment without having to sample each of the 288 plots, two sampling designs were applied to select 100 locations by taking into account spatial soil heterogeneity according to soil archive data [26]: 50 sampling points were selected according to stratified random sampling, and the other 50 sampling points were selected by employing the Kennard–Stone algorithm [27]. Plot margins of 1.5 m were excluded from sampling. Before measuring carbon with dry combustion, the soil samples were air-dried, sieved (2 mm), and powdered. Total carbon was assessed using the elemental analyzer, vario EL cube CN (Elementar Analysensysteme GmbH), with three replicates conducted for each sample. While carbonates were initially measured, their values were found to be inconsequential, and therefore were omitted from the analysis. As a result, the total carbon measured was



regarded as an indicator of SOC in this study. The observed SOC content is 19.6 g kg$^{-1}$ with a range of 14–25 g kg$^{-1}$, indicating a wide range of SOC values generated from different fertilization treatments.

Spectral measurements were made using a Veris® Vis-NIR spectrophotometer manufactured by Veris Technologies, Inc. (thus referred to as Veris). The Veris is equipped with an Ocean Optics USB4000 instrument (300 to 1100 nm) and a Hamamatsu TG series mini-spectrometer (1100 to 2200 nm), with a resolution of 4–6 nm. Due to logistical constraints, Veris field measurements were completed a year following soil collection in September 2019. Before conducting the Vis-NIR measurements, the soil's volumetric water content was assessed at each point location using a time domain reflectometry (TDR) moisture sensor. The TDR measurements indicated that the soil's moisture content ranged between 15 and 25%, signifying adequate soil moisture for establishing good soil contact of the sensor. The data were acquired at different dates to cover the entire field, and the soil water content at the moment of measurement was in the range of 15–30%. Several transects with a distance of 3–4 m were recorded covering the entire field and considering passing through the soil sampling points, obtaining about 10,000 data points (Figure 1). The spatial location of the on-the-go spectral measurements was initially recorded using the Veris instrument. These original GPS coordinates were then corrected and refined using a high-precision GNSS instrument, ensuring enhanced spatial accuracy and reliability in the positioning of the spectral measurements. Meanwhile, the spatial location of the soil sampling points was recorded directly using the GNSS instrument. The Veris spectrometer is built in a shank that is pulled through the soil by a tractor with a measurement depth of about 12 cm; measurements are taken through a sapphire window located on the shank's bottom. Approximately 20 spectra are captured each second [28]. The 400–2200 nm spectral range was used for model development.

### 2.3. Data Preprocessing

The PCOut function in the R-package mvoutlier was used to evaluate the soil spectra for outliers per LTE plot [29]. The scattering effects on the spectral signal were then reduced using various preprocessing approaches. The four combinations used were: Savitzky–Golay (SG; [30]), Savitzky–Golay + continuum removal (SGCR; [31]), Gap-Segment derivative (gapDer; [32]), and multiplicative scatter correction (MSC; [33]). Details are provided in Table 1. The prospectr R-package was used to obtain the SG, SGCR, and gapDer, and the pls R-package was used to obtain the MSC [34].

**Table 1.** Combinations of preprocessing techniques used in this study; w is window size, s is segment size.

| Preprocessing Method | Abbreviation | Veris Wavelength Range |
| --- | --- | --- |
| Savitzky–Golay | SG | 432–2201 |
| Savitzky–Golay w = 11 and continuum removal | SGCR | 432–2201 |
| Gap-Segment derivative (w = 11, s = 10) | gapDer | 408–2186 |
| Multiplicative scatter correction | MSC | 403–2201 |

### 2.4. Model Training and Evaluation

Model training was conducted in a two-step approach according to Figure 2. In Step 1, a regression model (R-model) is trained to relate the SOC content to the spectral information. In Step 2, the thereby obtained predictions of the SOC content are interpolated by ordinary kriging (K-model) to generate spatially continuous predictions throughout the area. We will refer to it as R + K modeling approach. It is not to be confused with regression kriging which would first build a regression model and then interpolate the residuals. Regression kriging would only be feasible if we have continuous spectral measurements throughout the area. However, this is not the case when on-the-go proximal sensing data are collected by sensors with a small spatial footprint.

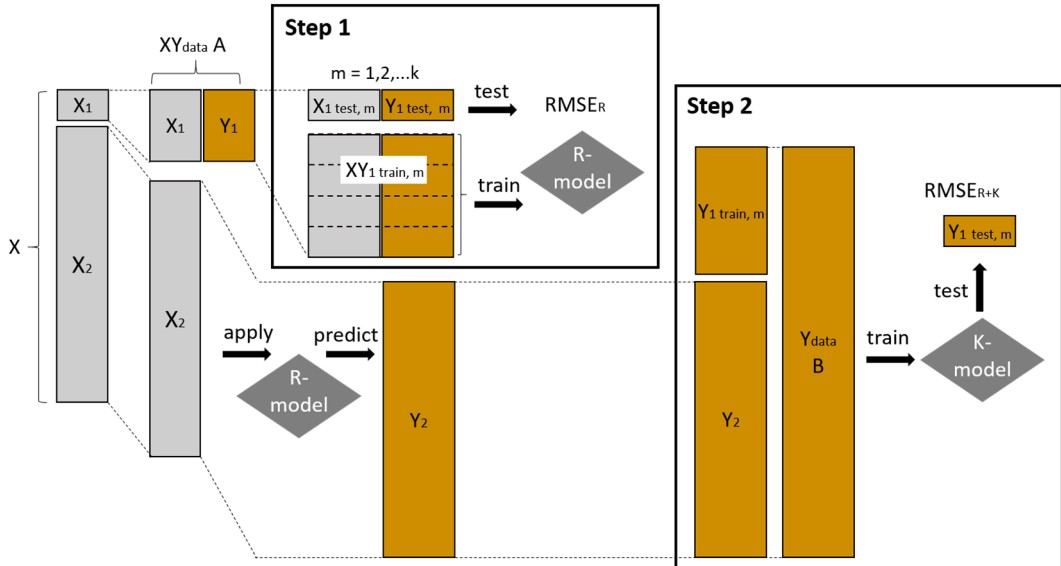

**Figure 2.** Two-step model training and evaluation procedure. Step 1: regression model training (R-model), Step 2: ordinary kriging (K-model). X = spectra, Y = SOC values, $RMSE_R$ = RMSE of the regression model, $RMSE_{R+K}$ = RMSE of the R + K modeling approach.

In our study, we chose to apply partial least squares regression (PLSR) to build the R-model. PLSR offers several advantages in dealing with high-dimensional data, effectively handling multicollinearity, and extracting relevant information from a large set of predictors. Additionally, PLSR is a well-established and interpretable method in the field of soil spectroscopy, providing meaningful insights into the relationship between soil spectra and soil properties [35]. With regards to Step 2, the K-model, one might argue, that simpler interpolation methods such as inverse distance weighting would also perform the job. However, only kriging allows for modeling spatial autocorrelation.

The R+K modeling procedure was implemented in the following way. First, the 10 spectral on-the-go measurements $X_1$ closest to each sampling point were averaged and assigned to the respective sampling point. Together with the average SOC value per sampling point, these data form the $XY_{data}$ A. In Step 1 of the modeling procedure, these data A are then used to train PLSR models by a nested *k*-fold cross-validation (CV) procedure. Each training set $XY_{train1,m}$ $with$ $m = 1, 2, \ldots k$ of the outer CV loop was again subdivided into *k*-folds in the inner CV loop to allow for model tuning, i.e., to determine the number of components. The PLSR model was then trained with $XY_{train1,m}$ and evaluated with $XY_{test1,m}$. After applying the respective PLSR model to all those spectra $X_2$, which were not assigned to any sampling point, the resulting SOC predictions $Y_2$ were combined with $Y_{train1,m}$ to form $Y_{data}$ B, the input data for Step 2 of the modeling procedure. In Step 2 of the modeling procedure, the $Y_{data}$ B was spatially stratified into *k*-folds making sure that each fold contained spectral measurement points from all LTE plots. This inner CV loop of modeling Step 2 was then used to determine the semivariogram parameters for ordinary kriging (OK). The K-models were again evaluated by the same test sets $Y_{test1,m}$ as the R-models.

Data subdivision for the nested CV accounted for possible spatial autocorrelation between training and test data in two aspects: (1) Nearby sampling points were assigned to the same fold, and (2) spectral measurements in the near surrounding of those spectral measurements were assigned to the sampling points to generate the $XY_{data}$ were excluded when building the K-model. The overall CV procedure was conducted with $k = 5$ and repeated five times, resulting in 25 R + K models and 25 spatially continuous predictions for each of the four differently preprocessed datasets and the three different semivariogram models: Spherical, Exponential, and Gaussian. Equal data subdivisions were used to allow

for direct comparison. The root mean square error (RMSE) was used to evaluate model performance. Spatially continuous predictions were realized with 1 m spatial resolution.

Due to the structure of the LTE (divided into plots with different treatments), the maximum spatial separation distance considered to construct the experimental variogram was 10 m. Alternative kriging approaches including pair of point aggregation and block kriging were also applied to pay tribute to data collection on an LTE. The PLSR models were trained with R-package pls, and the geospatial analysis was carried out using the R-package gstat [36,37]. The plots were created using the R-packages ggplot2 [38,39] and lattice [40].

## 3. Results and Discussion

### 3.1. Model Structure

Figure 3 shows distinct patterns in the optimal number of PLSR components for each preprocessing method. The SG and SGCR methods generally required a higher number of components, indicating a need to capture finer details and variations in the data. This aligns with the smoothing and denoising properties of the SG filter and the inclusion of continuum removal in the SGCR method to preserve intricate features. In contrast, the gapDer and MSC methods tended to require fewer components, suggesting a concise representation of the data. The gapDer method effectively reduced noise and identified important spectral regions through gap segmentation, while the MSC method corrected for multiplicative effects, enhancing the accuracy of the spectral information.

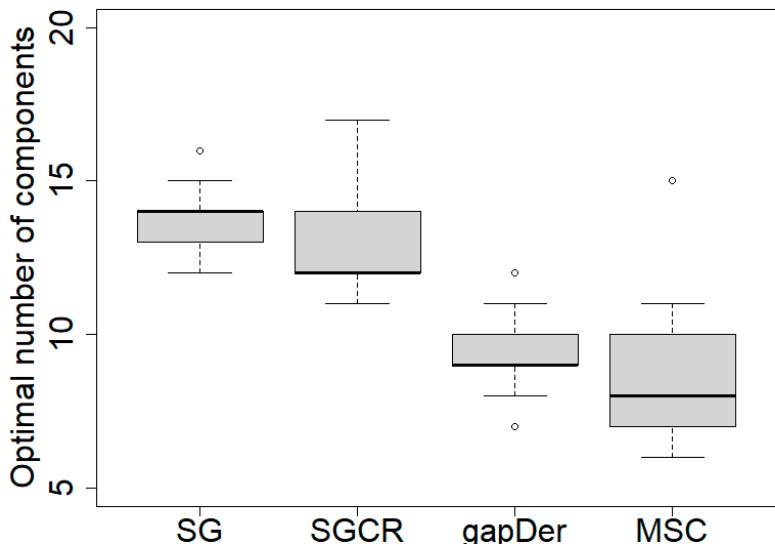

**Figure 3.** Boxplots of the optimal number of PLSR components of the 25 models trained with data preprocessed by SG: Savitzky–Golay, SGCR: Savitzky–Golay + continuum removal, gapDer: Gap-Segment derivative, and MSC: multiplicative scatter correction.

Figure 4 shows the K-models corresponding to the R-model predicted SOC of the on-the-go spectral data ($Y_{data}$ B). When comparing the K-models built from the predictions on behalf of the differently preprocessed data, there is a similarity in the spatial structure, although the semivariance in SG and gapDer is slightly lower. However, there is a difference in the parameter values between semivariogram models.

The Spherical model exhibited a smoother and more gradual change in the variable being measured, indicating a lower level of small-scale variability. This implies that neighboring data points within a certain distance tend to have similar values. In terms of maximum variability, the Spherical model displayed moderate to high levels, suggesting significant variations across the dataset. Furthermore, the Spherical model had larger spatial correlation ranges, indicating a wider extent of influence between data points.

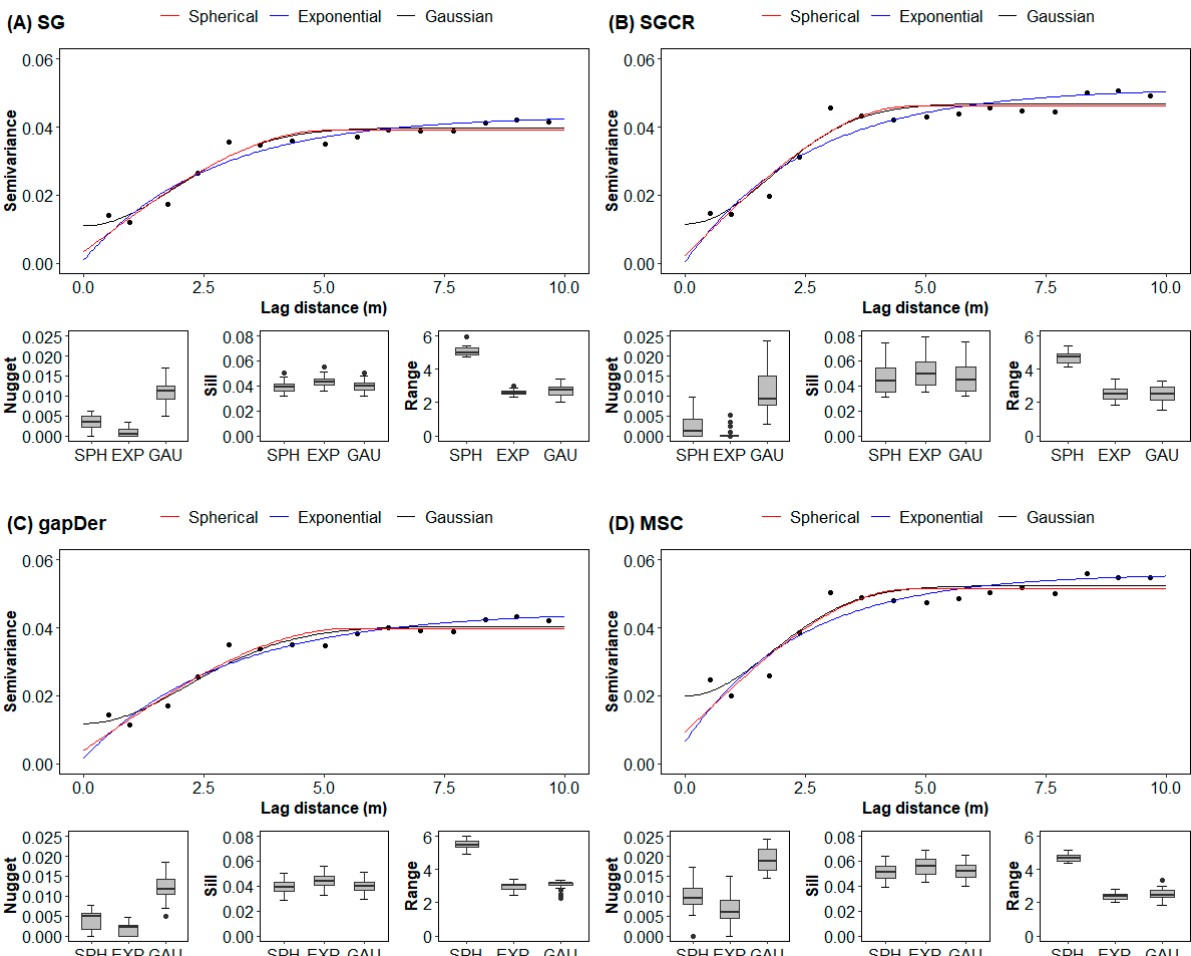

**Figure 4.** Semivariogram models corresponding to the PLSR predicted SOC of the on-the-go spectral data using the 25 models. The semivariogram model lines correspond to the average values, while the boxplots show the variation of the parameter values. (**A**) SG: Savitzky–Golay, (**B**) SGCR: Savitzky–Golay + continuum removal, (**C**) gapDer: Gap-Segment derivative, and (**D**) MSC: multiplicative scatter correction. SPH: Spherical, EXP: Exponential, GAU: Gaussian.

The Exponential model displayed moderate levels of small-scale variability, characterized by a decay pattern where nearby data points were more similar than those farther apart. Its spatial correlation range was generally smaller than that of the Spherical model, indicating a more rapid decrease in correlation with increasing distance.

The Gaussian model, however, exhibited similar patterns of small-scale and maximum variability to the Exponential model. It captured intermediate levels of small-scale variability, displaying a balance between the smoother Spherical model and the decay pattern of the Exponential model. The Gaussian model's spatial correlation range was smaller than both the Spherical and Exponential models, suggesting a more localized influence of neighboring points. This suggests that data points which are nearby have a stronger impact on each other, while the impact diminishes rapidly as the distance increases.

### 3.2. Performance Metrics of PLSR and OK

Figure 5 presents the predictive model performance of the PLSR models. The best models were the ones using SG and gapDer with a median RMSE$_R$ value below 1.6 g kg$^{-1}$. They also indicate a lower dispersion compared to the other two models as is observable in the comparison of predicted versus measured values (Figure 6). SG is a common method that mainly smoothes the original signal to remove multiplicative and additive effects [41]. Meanwhile, the gapDer method works by derivate specific segments of the signal [42]. The

gapDer is the preprocessing method with a lower number of wavelengths used compared with the others selected in this study, thus the reduction in the model complexity [43] could have a positive effect in this case. The differences observed between preprocessing methods remark on the importance of selecting an adequate method. There is no standard procedure even under laboratory conditions since the required type and amount of preprocessing is data specific for soil [3,44].

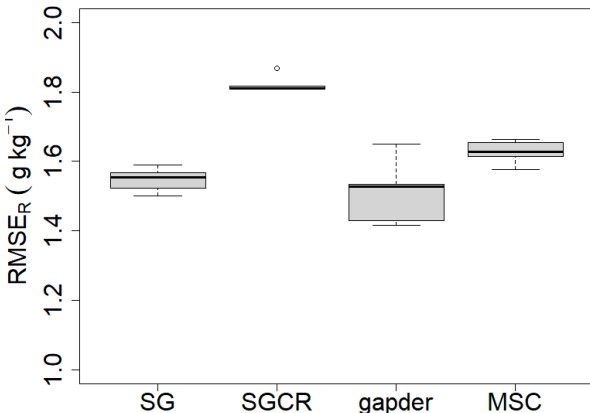

**Figure 5.** Predictive model performance of Step 1 for each preprocessing method (five values per boxplot). SG: Savitzky–Golay, SGCR: Savitzky–Golay + continuum removal, gapDer: Gap-Segment derivative, MSC: multiplicative scatter correction.

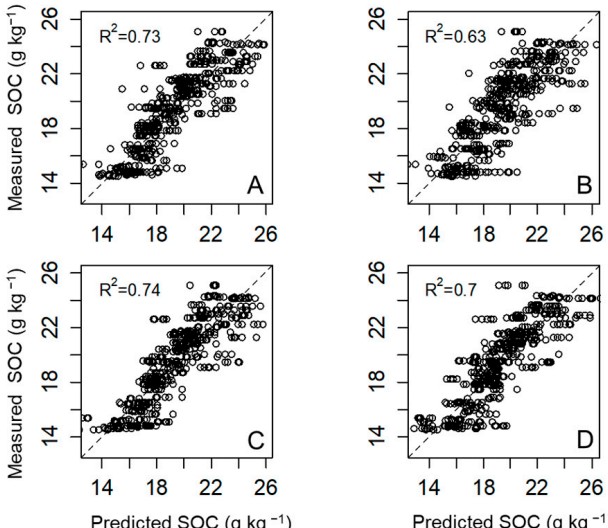

**Figure 6.** Predicted versus observed values of modeling Step 1, with five predictions per sampling location. (**A**) SG: Savitzky–Golay, (**B**) SGCR: Savitzky–Golay + continuum removal, (**C**) gapDer: Gap-Segment derivative, (**D**) MSC: multiplicative scatter correction.

The overall predictive model performance of modeling Step 1 + Step 2 (R+K model) is presented in Figure 7 and the scatter plot with a line of equality is shown in Figure 8. By including modeling Step 2, the overall predictive performance was further improved. The best predictive performance for Step 2 was achieved with the Gaussian model. Accordingly, the best results were obtained with the combination SG—Gaussian ($RMSE_{R+K}$ = 1.24 g kg$^{-1}$, $R^2_{R+K}$ = 0.84) and gapDer—Gaussian ($RMSE_{R+K}$ = 1.26 g k$^{-1}$, $R^2_{R+K}$ = 0.82). The observable pattern of the dispersion in the predictions (Figure 8) has more similarities concerning the preprocessing method than with regard to the semivariogram model.

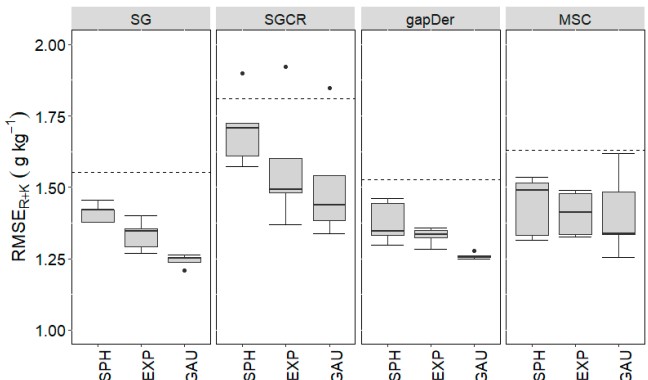

**Figure 7.** Overall predictive model performance (Step 1 + Step 2) for each preprocessing method and each semivariogram model. Dashed lines represent the median value obtained in Step 1. SG: Savitzky–Golay, SGCR: Savitzky–Golay + continuum removal, gapDer: Gap-Segment derivative, MSC: multiplicative scatter correction, SPH: Spherical, EXP: Exponential, GAU: Gaussian.

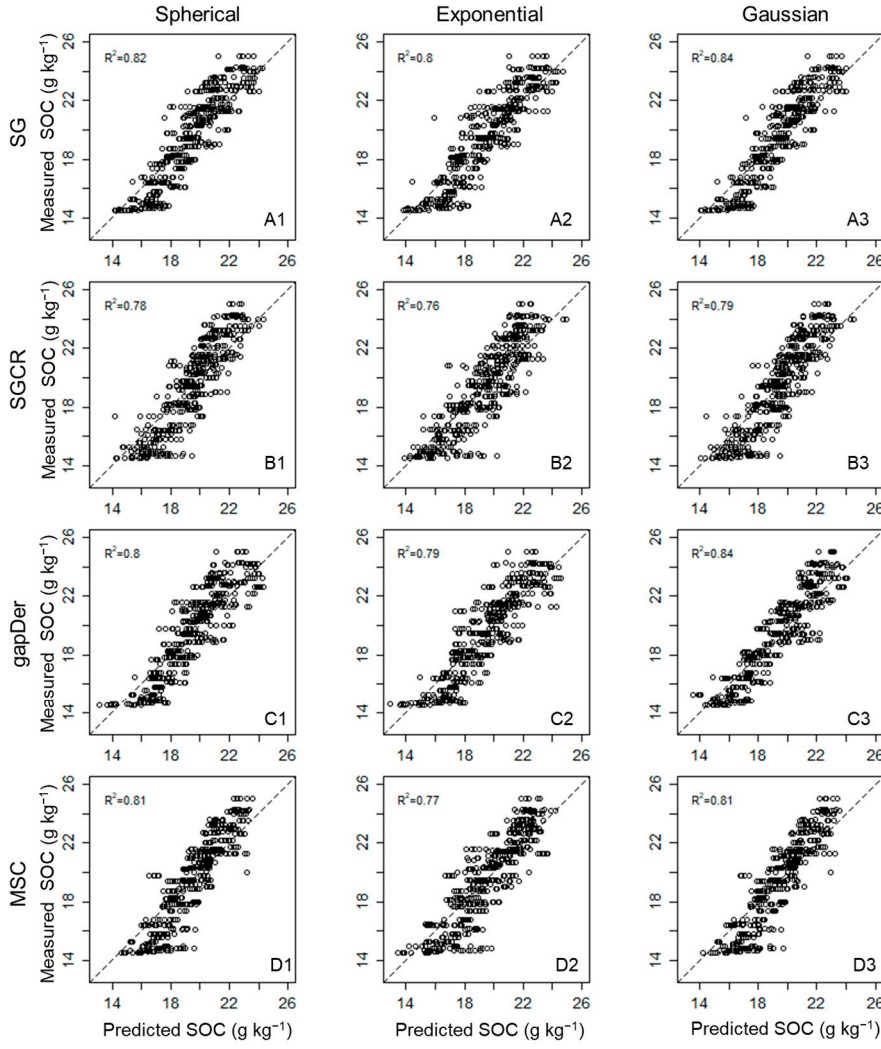

**Figure 8.** Predicted versus observed values of the overall modeling procedure (Step 1 + Step 2) with five predictions per sampling location. (**A**) SG: Savitzky–Golay, (**B**) SGCR: Savitzky–Golay + continuum removal, (**C**) gapDer: Gap-Segment derivative, and (**D**) MSC: multiplicative scatter correction. (**1**): Spherical, (**2**): Exponential, (**3**): Gaussian.

Another aspect to consider is the spatial distribution of the residuals. Figure 9 presents an example of the SG—Gaussian and gapDer—Gaussian methods. Both approaches present similarities in the distribution of the residuals, and the majority is in the range of $-0.5$–$0.5$ g kg$^{-1}$. There is no clear trend based on the plot size or the cluster division used for the sampling design, although areas with higher SOC showed higher interquartile range values and vice versa in the case of areas with low SOC values.

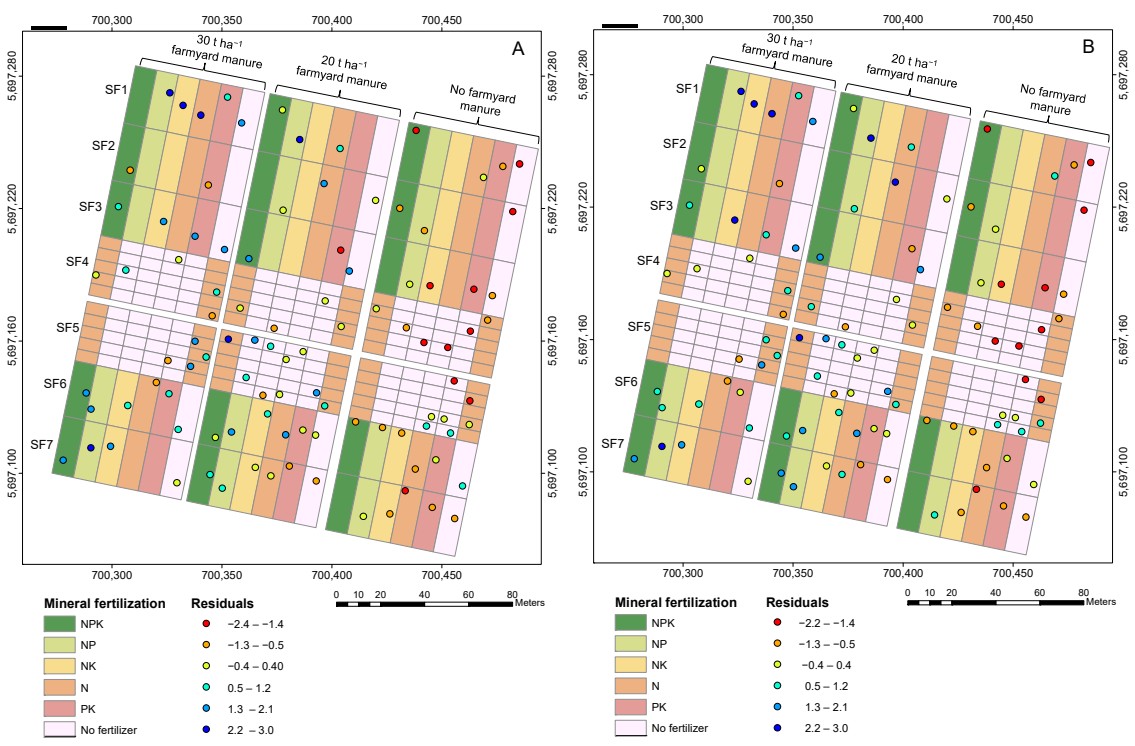

**Figure 9.** Average residuals of predicted SOC values per sampling point location of (**A**) Savitzky–Golay—Gaussian, and (**B**) Gap-Segment derivative—Gaussian methods.

### 3.3. Spatially Continuous Predictions of SOC

The spatially continuous prediction of SOC presented similar patterns independent of the combination of methods used for interpolation. As an illustration, Figure 10 presents the maps of the estimated SOC using the models with the best performance (SG—Gaussian and gapDer—Gaussian) and the difference in prediction between them. The SOC values presented a range of about 10–30 g kg$^{-1}$, which is a wider range compared with the laboratory samples (14–25 g kg$^{-1}$). Not only the pattern is similar between methods but also the differences in the prediction were low with the exception of some specific areas. Figure 11 displays the spatial predictions of the same models comparing the interquartile range distribution of 25 predictions for each one. In general, the median interquartile range is below 1 g kg$^{-1}$ in both cases. Altogether, the spatial variation is most homogeneous in the case of the SG—Gaussian model.

Predictive uncertainty decreased with the combined use of the R and K models. This improvement can be attributed to the similarity in spectral data among neighboring points located within the same treatment plot, indicating lower soil variation within the plot. Among the semivariogram models, the Gaussian model exhibited effective balancing of small-scale variability and spatial correlation. It displayed a rapid decrease in correlation with distance, emphasizing localized influences, and displayed a lower level of small-scale variability. These characteristics resulted in better predictions compared to the Spherical and Exponential semivariogram models (Figure 4). It is worth noting that the maximum spatial separation distance used for the semivariogram model in this study was short (10 m), focusing on capturing influences within the plot. However, the effectiveness of

the model may vary throughout the area due to differences in LTE plot sizes. Therefore, selecting a model that accurately represents the spatial structure is crucial for reliable interpolation [45]. Notably, when examining the residuals, similar trends were observed across different combinations, with higher values tending to be underestimated and lower values tending to be overestimated.

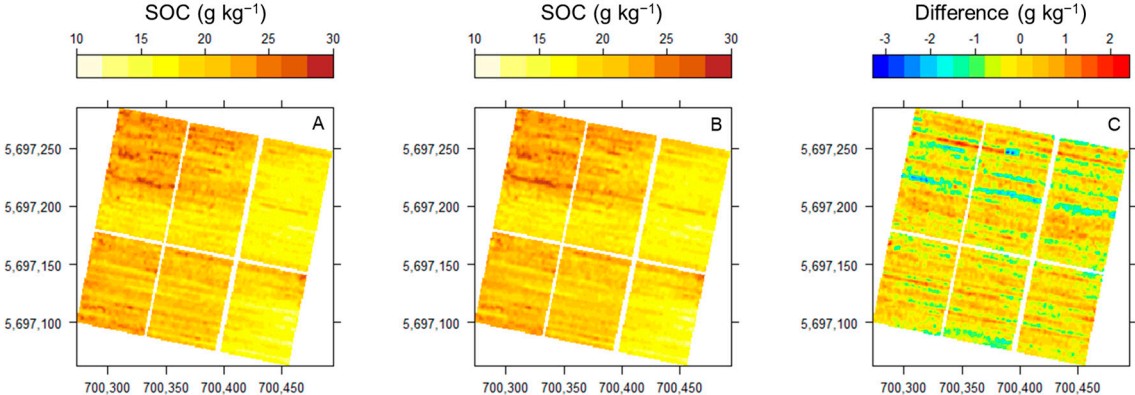

**Figure 10.** The median of predicted SOC values of the R+K models with the best performance. (**A**) Savitzky–Golay—Gaussian, (**B**) Gap-Segment derivative—Gaussian, (**C**) difference between models.

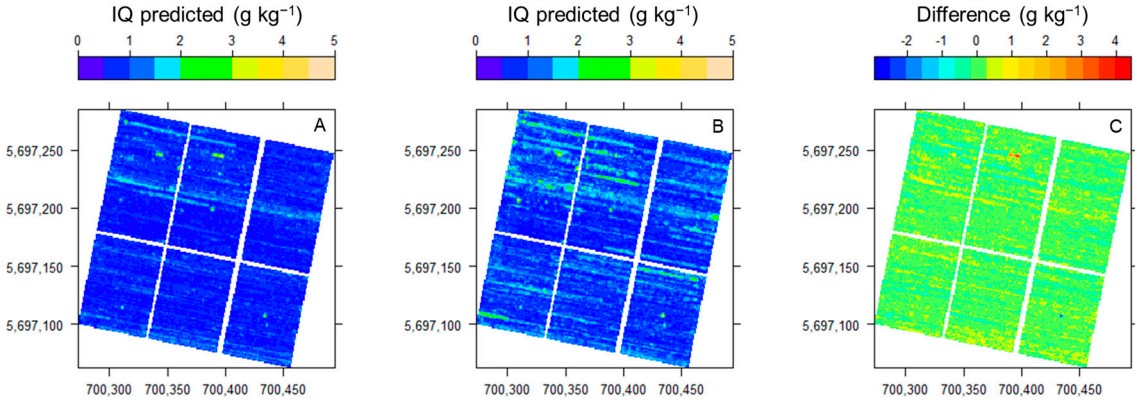

**Figure 11.** Interquartile range (IQ) of predicted SOC values of models with the best performance and the difference between predictions. (**A**) Savitzky–Golay—Gaussian, (**B**) Gap-Segment derivative—Gaussian, (**C**) difference between models.

Previous studies used in situ soil spectral measurements, e.g., [46,47], and some of them have used a Veris spectrometer [4,11,12,28,43]. The model performance in our study presented better RMSE values to predict SOC compared with these studies (best RMSE = 2.7 g kg$^{-1}$), although our $R^2$ was lower compared with [43] ($R^2$ = 0.90). While the comparison is not straightforward due to differences in the SOC range, field conditions, and model evaluation procedure, and no other study has used an on-the-go spectrometer on an LTE. Our results presented high accuracy showing the high potential of our approach for field-scale SOC monitoring.

Regarding the generated maps, different methodological data processing and modeling combinations resulted in similar SOC spatial distribution, which could be expected due to the high sampling density of the Veris measurements reducing the uncertainty of the spatial interpolation. A striping effect in the SOC distribution maps was evident, which was likely caused due to the Veris transect measurements in one direction [12] and due to the short maximum spatial separation distance considered for the experimental semivariograms. This effect could be changed with data aggregation using different approaches. To illustrate alternatives for mapping the field, Figure 12 presents maps using a

Savitzky–Golay—Gaussian model with pair of point aggregation (Figure 12A), another with extending the maximum spatial separation distance of the experimental semivariogram to 25 m (Figure 12B), and using block kriging with blocks defined by the plot treatments (Figure 12C). By pair of point aggregation, the striping effect is diminished but still visible. When extending the maximum spatial separation distance, the striping effect disappears, and a more general SOC distribution is observed. Nevertheless, a generalization of the SOC distribution could mask the values of small plots and the spatial variation inside the plots; therefore, it is better applied in fields with homogeneous management. In the case of block kriging, a map with blocks divided by the plot treatments is presented. Block kriging has been less used in soil mapping compared with point kriging methods [48]. Generally, it uses blocks of the same size to upscale point observations [49]. In the LTE, the block kriging approach could be an alternative to monitor SOC changes by having a unique value per plot treatment, although it will not represent the internal variation inside the plot.

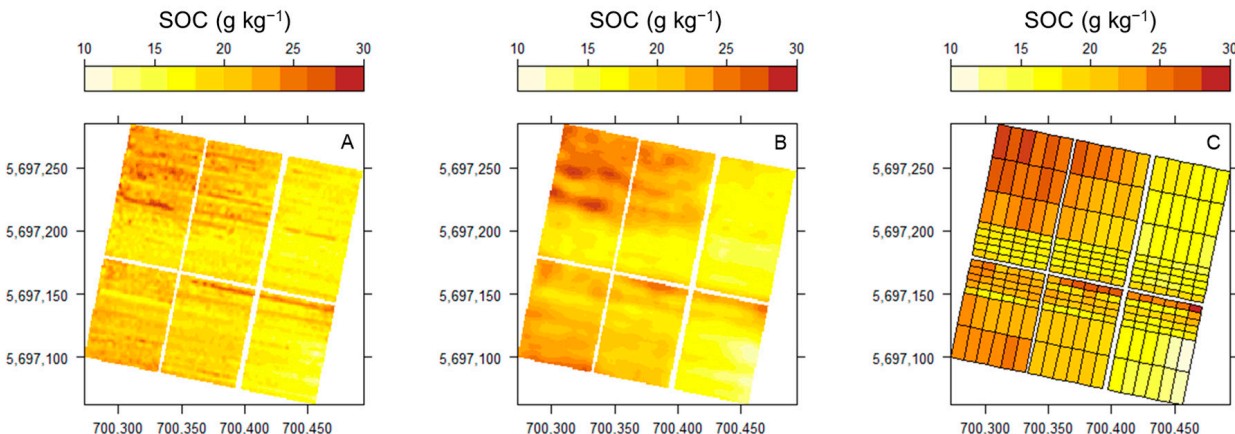

**Figure 12.** Maps of the median of predicted SOC values using the Savitzky–Golay—Gaussian method using three different approaches, (**A**) pairing points of Veris data, (**B**) extending the maximum spatial separation distance for the semivariogram to 25 m, (**C**) applying block kriging with the field plots as block delineation.

Mapping SOC has also been studied with remote sensing using airborne and satellite platforms to cover extended areas although with lower precision. Consequently, it should be integrated with field and laboratory measurements and complementary sensor data for better results [50]. Our results showed the feasibility of using on-the-go soil spectra for mapping SOC with appropriate reliability, having an accuracy closer to laboratory measurements than remote sensing data. Overall, different challenges appear when using field measurements due to environmental factors [17]. For example, peaks in the spectral signal associated with soil water content can obscure peaks related to organic functional groups [12]. Different methods have been tested to correct for disturbance effects impacting sensor measurements under field conditions [51,52]. In the context of on-the-go Vis-NIR spectroscopy for SOC monitoring, it is crucial to consider the potential effect of temporal variation on spectral data. While our study successfully estimated SOC levels with high spatial resolution, it is essential to acknowledge that agricultural SOC levels can exhibit temporal variability due to various factors. These include seasonal changes, crop rotations, management practices, natural disturbances, and soil moisture fluctuations. These temporal variations can introduce additional complexity to the spectral data and may influence the accuracy of predictive models over time. To address this, continuous monitoring of SOC levels through multiple on-the-go measurements at different time points are required. Long-term monitoring can help to identify trends and seasonal patterns, providing valuable insights into SOC dynamics under varying environmental conditions.

## 4. Conclusions

The prediction of SOC using on-the-go field spectra demonstrated promising results. PLSR models, constructed with spectra close to the sampling location, effectively predicted the remaining Veris measurements, enabling the creation of high-resolution field maps. An enhancement in model performance was evident when PLSR was synergized with ordinary kriging (R+K model) to generate continuous predictions, making this combination particularly noteworthy.

The preprocessing methods Savitzky–Golay (SG) and Gap-Segment derivative (gapDer) stood out for their efficacy, and this was further accentuated when paired with a Gaussian semivariogram model. The boost in model performance upon utilizing these methods suggests that there is an inherent similarity in spectral data among neighboring areas and within identical treatment plots. This improvement emphasizes the potential significance of these techniques in efficiently capturing spatial patterns and dependencies in the context of SOC prediction.

When the different R+K model predictions were compared, a pronounced similarity in the spatial distribution of SOC was observed, which is consistent with the expectations due to the high-density data collected using Veris. Nonetheless, the striping effect became apparent due to the data being gathered in transects and the use of a relatively small maximum spatial separation distance for the semivariograms. Alleviating this striping effect could be achieved by extending the spatial separation distance, employing data aggregation techniques, or defining the distribution based on treatment plots (i.e., block kriging or similar methods). The applicability of data aggregation is contingent on the layout of the field and the specificity of the information sought. It is crucial to acknowledge that employing field soil spectroscopy for predicting SOC at field scale is an area still in development. However, the results of this study underline the potential of this technique in the continuous monitoring of SOC. There is an imperative need for ongoing efforts to refine and establish standard practices for spectral soil measurements under field conditions.

**Author Contributions:** Conceptualization, M.L.; formal analysis, J.R. and M.L.; funding acquisition, M.L.; investigation, J.R. and M.L.; project administration, M.L.; supervision, M.L.; writing—original draft, J.R.; writing—review and editing, M.L. All authors have read and agreed to the published version of the manuscript.

**Funding:** This work was supported by funds of the Federal Ministry of Food and Agriculture (BMEL) based on a decision of the Parliament of the Federal Republic of Germany via the Federal Office for Agriculture and Food (BLE) under the innovation support program. Project SOCmonit–Monitoring of soil organic carbon with remote and proximal soil sensing methods (grant number 281B301516).

**Institutional Review Board Statement:** Not applicable.

**Data Availability Statement:** The data will be made available as part of the R package SOCmonit.

**Conflicts of Interest:** The authors declare no conflict of interest.

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
