# Peer review of "On-the-Go Vis-NIR Spectroscopy for Field-Scale Spatial-Temporal Monitoring of Soil Organic Carbon"

_agriculture, doi:10.3390/agriculture13081611_

Round 1

Reviewer 1 Report

Very good day

The manuscript made available is of scientific relevance, applicable to agriculture.

work must be done to homogenize the way of writing the units of measurement

In the introduction, the citations must be updated, since the state of the art in the subject must be clear and the reason for the research based on its beginnings and to what point it is at the moment so that the reader is clear about the reason and the importance of research

In the introduction there are extremely long paragraphs (approximately 36-38 lines) that make understanding a bit difficult. I suggest these may have between 10 to 15 lines, the ideas may be more specific and better understood.

This recommendation is for the entire document.

In materials and methods, section 2.1 I suggest analyzing the title because it not only refers to the study area but also to the conditions and its management, so it should be titled study area. Conditions and management

Results and discussion In the case of figure 4, the graphs that appear with the parameters of the models do not have a homogeneous size, this must be corrected.

Many results and very little discussion of these, the discussion of the results should be deepened, and use more current references from the last 5 years

Reviewer 2 Report

Dear authors,

I have enjoyed your reading your manuscript. It is interesting and you obtained some good results. I know that it is not easy with the mobile platform.

I have some comments to improve the manuscript.

The temporal part of the monitoring is not developed enough in the Discussion.

First of all, there are many typos and the language in the Introduction could be improved. 

The reference list is not formatted according to the Journal format.

Some references listed are not cited in the manuscript. On the other hand, some other are in the manuscript but not in the list.

You could also add some more references in the Introduction section.

Abstract

L18: give the signification of the abbreviation (SG) – Savitzky and Golay

L23: immense, it is quite a strong word!

Introduction

L32: what do you mean by data-driven soil health? In that sentence, some references are needed (L31-33). 

L38: granular level. What do you mean?

L46: in the VIS-NIR field (especially on-the-go), ancillary data is rather a term used for data supporting the spectra during the modelling (moisture, temperature, etc). You should reformulate the sentence. 

L50: the method is not so new

L51: amass sounds weird here.

L53: ref missing in the list

L58: the utilization of on-the-go sensors is still very difficult / cumbersome in my experience with the Veris system!

L58: also a ref missing here

Vis-NIR, VisNIR: chose one form and keep it all along

L74-76: … among the chemometric methods…

PLSR is rather a chemometric method not an analytical on. On the other hand, VIS-NIR spectroscopy is an analytical method.

L87: add here some factors for the reader not aware of the method.

L92-93: reformulate

Materials and Methods

L112: not sure to find the reference in the list

Maybe change the name of section 2.1 to Study area and LTE

L136: ref is missing

L189: you mean averaging the 10 on-the-go spectra nearest to the sampling point?

Results and Discussion

L351: ref missing

Overall, the Discussion is interesting!

You should look for the several typos in the text and improve the English in the Introduction.

Reviewer 3 Report

Overall a valid contribution to the field. The authors do need to justify their analysis decisions more in the manuscript. Specifically, why they chose to use PLSR and ordinary Kriging. There are plenty of other machine learning models that have outperformed both for the spectroscopy and spatial interpolation space. That said there are still valid reasons to use PLSR and OK, I believe the authors need to articulate why as compared to using more complex modelling approaches. 

The other issue is the authors need to add more detail about the variability of moisture contents in the samples. Specifically how consistent it was. Soil water content is a major issue for field spectroscopy, and generalizing across water contents is difficult. The authors need to discuss that more, and include that in their interpretation of the results. 

Line 179: Spectroscopy literature has shown value of using a range of other model types (i.e. Cubist) compared to PLSR. There are still advantages to PLSR in some situations, but machine learning models have often outperformed PLSR. The authors should justify why they used PLSR.

Line 183: Why as ordinary kriging used compared to using predictive soil mapping techniques?

Line 365: Recommend authors try and account for anisotropy to try and deal with striping

Line 151: More detail on soil water content, and how consistent it was across the samples. Major issue with field spectroscopy.

Reviewer 4 Report

The paper presents a comprehensive investigation on the spatial prediction of SOC content on field level using the on-the-go spectrophotometer and different statistical procedures.  The object, design and realization of the investigarion is very good.

Some technical errors and minor Emglish editing is needed (e.g. rows 28-30, 44-45, 410),  The statement in rows 83-85 repeats the text in 63+65.

Round 2

Reviewer 2 Report

Dear authors,

The manuscript was improved. Thank you.

I have no other remarks.

In some of the reworked sections, the English can sometimes be improve (e.g. L70: avenue >> venue; L253-254: colloquial language...).

Reviewer 3 Report

Authors have addressed the review comments.